# Dietary Acid Load in Gluten-Free Diets: Results from a Cross-Sectional Study

**DOI:** 10.3390/nu14153067

**Published:** 2022-07-26

**Authors:** Maximilian Andreas Storz, Alvaro Luis Ronco, Mauro Lombardo

**Affiliations:** 1Department of Internal Medicine II, Center for Complementary Medicine, Freiburg University Hospital, Faculty of Medicine, University of Freiburg, 79106 Freiburg, Germany; 2Unit of Oncology and Radiotherapy, Pereira Rossell Women’s Hospital, Bvard. Artigas 1590, Montevideo 11600, Uruguay; alv.ronco58@gmail.com; 3Department of Human Sciences and Promotion of the Quality of Life, San Raffaele Roma Open University, 00166 Rome, Italy; mauro.lombardo@uniroma5.it

**Keywords:** dietary acid load, potential renal acid load, net endogenous acid production, gluten-free diet, celiac disease, nutritional epidemiology, grains, acid–base homeostasis

## Abstract

The gluten-free diet (GFD) ensures improvement of clinical symptoms in the vast majority of celiac disease (CD) patients. Despite stable CD rates in many countries, an increasing number of healthy individuals are adopting gluten-free diets, believing that this diet is an inherently healthier choice. The health effects of gluten-free diets are controversial, and a recent study added to the debate by reporting a lower acidogenic potential of this diet. The effects of the GFD on potential renal acid load (PRAL) and net endogenous acid production (NEAP)—two important markers of dietary acid load (DAL)—are poorly understood, and have never been examined in a Western population. Using cross-sectional data from the National Health and Nutrition Examination Surveys, we estimated DAL in U.S. individuals reporting a GFD and contrasted the results to the general U.S. population consuming gluten and denying special diets. The GFD was associated with significantly lower crude DAL scores, and after adjustments for confounders in multivariate regression, the results remain significant. Yet, our study could not confirm the reported alkalizing properties of the GFD. Although overall DAL scores were significantly lower in the GFD group, they were comparable to Western diets producing 50–75 mEq of acid per day.

## 1. Introduction

In nutritional science, the term gluten refers to certain cereal prolamins, including the ethanol-soluble proteins of rye, wheat, spelt, kamut and barley [1,2]. Gluten is a high-molecular-weight seed storage protein and an integral component of wheat-containing staple foods such as pasta, bread and cereals [2,3,4].

In individuals with celiac disease (CD), gluten regularly triggers an autoinflammatory process in the intestinal mucosa, resulting in an immune-mediated enteropathy [4,5]. Classical symptoms of CD include malabsorption, recurrent abdominal pain and chronic diarrhea [5]. Recent studies also demonstrated an increasing rate of non-classical and subclinical phenotypes [6], highlighting the wide range of extraintestinal disease manifestations [6,7].

A gluten-free diet (GFD) ensures improvement of clinical symptoms and signs of malabsorption in the vast majority of CD patients [8]. However, despite stable CD rates in many countries, the popularity of gluten-free diets has risen steadily in recent decades [4]. A study from New Zealand suggested that approximately 1% of children suffered from CD, but almost 5% reported gluten avoidance [9]. A study in American adults revealed a comparable picture, and reported a significant overall increase in individuals adhering to a GFD [10].

Many individuals that do not suffer from CD mistakenly believe that a GFD is an inherently healthier choice, and may be beneficial with regard to non-specific gastrointestinal symptoms [4]. Thus, a GFD is oftentimes adopted for no medical reasons. Notably, several studies reported nutritional deficiencies in individuals following a GFD [11,12]. Gluten-free diets were characterized by a higher saturated fat content and an increased glycemic index [12].

Gluten-free diets are not recommended for healthy individuals. The health effects of GFDs in individuals without CD are controversial and subject to ongoing research. In this context, Nikniaz et al., recently reported a lower acidogenic potential of gluten-free diets in an Iranian population of CD patients [13]. Potential renal acid load (PRAL) and net endogenous acid production (NEAP)—two important markers of dietary acid load (DAL)—were significantly lower in individuals following a GFD as compared to the general population.

However, the authors neither reported total energy intake nor protein intake [13]. Both are important determinants of DAL and necessary to understand the effects of a particular diet on acid–base homeostasis [14,15,16]. Replacing cereals with other cereals per se does hypothetically not affect DAL (Table 1). Substituting 100 g of wheat with 100 g of millet has, for example, a negligible impact on the total PRAL score (+0.4 mEq/day).

The very low PRAL scores in the GFD group reported by Nikniaz et al., thus warrant further investigation [13]. Whether they were the consequence of a lower total protein intake (or due to another specific dietary feature) was not ascertainable from the published data [13]. Thus, there remain many open questions and—to the best of our knowledge—DAL has never been investigated in a Western population consuming a GFD.

The present analysis sought to address this gap in the literature, using aggregated cross-sectional data from the National Health and Nutrition Examination Survey (NHANES). The NHANES is an ongoing nationally representative project designed to assess the general health and nutritional status of children and adults in the United States. We used parts of the collected data to estimate DAL and nutrient intake in U.S. individuals reporting a GFD. The results were then contrasted to the U.S. general population consuming gluten products and denying a special diet (e.g., low-carbohydrate diets, vegetarian diets, etc.).

## 2. Materials and Methods

### 2.1. The NHANES

The NHANES is a national health-related program conducted by the Centers for Disease Control and Prevention’s National Center for Health Statistics in the United States [17]. Background data on the NHANES have been described elsewhere in great detail [18]. In brief, the NHANES is a large and ongoing project designed to assess the general health and nutritional status of the U.S. civilian population. NHANES collects data on demographics, dietary intake and other health behaviors and was designed to represent the total civilian non-institutionalized population in the United States [19,20]. Approximately 5000 people are enrolled per year. A key NHANES feature is the multistage probability sampling procedure. Data are collected by a specifically trained team of interdisciplinary interviewers. NHANES data have been used multiple times in the past to investigate scientific questions about the GFD and CD [10,21,22].

For this retrospective analysis, we used data from various NHANES modules, including demographic data, dietary data, anthropometric data and data from the medical conditions section. Three continuous NHANES cycles were appended to increase the sample size for analysis (2009/2010, 2011/2012 and 2013/2014). The analysis was performed between May and June 2022.

### 2.2. Dietary Acid Load Calculations

The employed methods for DAL estimations have been discussed elsewhere in detail [16]. In brief, we used 3 established formulas to calculate DAL from daily nutrient intake: PRAL_R_ (potential renal acid load—based on the Remer formula), NEAP_R_ (net endogenous acid production based on the Remer formula) and NEAP_F_ (net endogenous acid production based on the Frassetto formula). We calculated the PRAL_R_ of diet as follows [23]:PRAL_R_ (mEq/day) = (0.49 × total protein (g/day)) + (0.037 × phosphorus (mg/day)) − (0.021 × potassium (mg/day)) − (0.026 × magnesium (mg/day)) − (0.013 × calcium (mg/day)

NEAP_F_ was calculated as follows [24]:NEAP_F_ (mEq/day) = (54.4 × protein (g/day)/potassium (mEq/day)) − 10.2

Finally, we estimated NEAP_R_ based on the PRAL score and anthropometry-based estimates for organic acid excretion (OA_est_):NEAP_R_ (mEq/day) = PRAL_R_ (mEq/day) + OA_est_ (mEq/day)
where OA_est_ (mEq/day) was calculated as follows:OA_est_ (mEq/day) = [0.007184 × height(cm)^0.725^ × weight(kg)^0.425^] × 41/1.73

We refer the reader to the work of Parmenter et al., for additional background information on all scores [14,25].

### 2.3. Nutrient Intake Assessment

We obtained dietary data from the NHANES dietary interview component. The nutritional assessment component of the NHANES was conducted in person by trained dietary interviewers fluent in Spanish and English. Data were based on a computerized 24 h dietary recall method to estimate energy and nutrient intake for all participants. The examination protocol as well as the data collection methods are fully documented in the NHANES dietary interviewer’s procedure manuals [26]. Additional information may be obtained from our previous publication [18].

### 2.4. Dietary Pattern Assessment

Adherence to a GFD was assessed based on data from the medical conditions section, which provides self-reported personal interview data on a broad range of health conditions. Participants were asked “Are you on a gluten-free diet?”. We considered only individuals that gave a definite answer (“yes” or “no”) and excluded the remaining participants. Individuals who denied consumption of a special diet and consumed gluten-containing foods were used as a “control group”. Allocation was based on the question “Are you currently on any kind of diet, either to lose weight or for some other health-related reason?”. Again, only individuals that gave a definite answer (“yes” or “no”) were considered.

### 2.5. Inclusion and Exclusion Criteria

The adult study population aged 20 years or older from the 3 aforementioned NHANES cycles (2009/2010, 2011/2012 and 2013/2014) was analyzed in this study. We only considered participants with a full dataset for the present analysis. No data imputation took place, and participants with missing data on any study characteristics were excluded. Individuals with an implausible energy intake (e.g., a total energy intake ≤800 kcal/d or an energy intake ≥5000 kcal/d) were excluded. We also excluded individuals with implausible DAL scores.

### 2.6. Statistical Analysis

We used STATA 14 statistical software (StataCorp. 2015. Stata Statistical Software: Release 14. College Station, TX, United States) for this analysis. In the first step, we constructed appropriate sample weights to account for the complex, multistage, probability sampling design of the NHANES. In the second step, we investigated whether both compared groups (individuals on a GFD vs. individuals who did not report a special diet) had a different total energy intake. The significantly lower total energy intake in individuals on a GFD prompted us to employ a commonly used energy adjustment method (reporting nutrient intake in gram or milligram/1000 kcal) in addition to reporting total intakes [27].

Categorical variables were described with the unweighted number of observations and the weighted proportions in parentheses. Unreliable weighted proportions were identified using Stata’s post-estimation command “kg_nchs”, and those not meeting the NCHS standards were clearly flagged with superscript letters.

We used histograms, box plots and subpopulation summary statistics to check for frequency distribution and normality of the data. Normally distributed variables were described with their mean and standard deviation. STATA’s Rao–Scott test (a design-adjusted version of the Pearson chi-square test) was used to explore potential associations between dietary group and the respective variables.

Student’s *t*-test was used to compare intergroup differences in macro- and micronutrients and DAL scores. Furthermore, we used linear regression analyses to examine the relationship between all 3 DAL scores (NEAP_F_, PRAL_R_, NEAP_R_) and a selected set of independent variables. Finally, we used marginsplots to graph statistics from fitted models.

Statistical significance was determined at α = 0.05, and all tests for statistical significance were two-sided.

## 3. Results

The final study sample after exclusion of participants with missing data comprised *n* = 12,439 individuals, of which *n* = 187 individuals reported consumption of a GFD. Our total sample might be extrapolated to represent 179,491,354 U.S. Americans. Figure 1 shows the participant inclusion flowchart for the present analysis.

We observed no differences with regard to age when comparing both groups. The weighted proportion of females reporting a GFD was significantly higher. Table 2 displays additional sample characteristics.

No statistical intergroup differences were found with regard to marital status and race/ethnicity. Notably, the weighted proportion of individuals with a college degree (or above) reporting a GFD was significantly higher as compared to those denying a special diet.

Table 3 displays energy intake as well as macro- and micronutrient intake in the investigated sample. Significant intergroup differences were found with regard to total energy intake, carbohydrate intake and magnesium intake. After adjusting for total energy intake, we observed a significantly higher intake of potassium, magnesium and phosphorus per 1000 kcal in the GFD group. Mean total protein intake was lower in individuals reporting a GFD; however, the intergroup difference was not statistically significant.

Table 4 shows crude DAL scores across both groups. Both groups consumed an acidogenic diet (as shown by the total mean PRAL_R_ values >0 mEq/day). All three crude DAL scores were significantly lower in the GFD group.

Multivariate regression models revealed significantly lower PRAL_R_ values in the GFD group after adjustment for confounders. The difference was more pronounced in model 2, in which we also adjusted for protein intake (Table 5). A significant regression equation was found for model 1 (F(9,40) = 106.60, *p* < 0.001), with an R² value of 0.1940, and for model 2 (F(10,39) = 367.15, *p* < 0.001), with an R² value of 0.4536.

There were no differences in NEAP_F_ between either group in model 1. However, when adjusting for protein intake, the intergroup difference became statistically significant (Table 6). Again, significant regression equations were found for model 1 (F(9,40) = 91.18, *p* < 0.001), with an R² value of 0.0848, and for model 2 (F(10,39) = 149.55, *p* < 0.001), with an R² value of 0.2549.

Based on model 2 for PRAL_R_, Figure 2 shows the marginal predicted values of PRAL_R_ for both diet groups (GFD in red, no special diet in blue) at all possible increments of 10 units in age. Similarly, based on model 2 for NEAP_F_, Figure 3 shows the marginal predicted values of NEAP_F_ for both diet groups (GFD in red, no special diet in blue) at all possible increments of 10 units in age.

## 4. Discussion

Our data suggest a lower acidogenic potential of GFDs in the examined sample. Crude DAL scores were significantly lower in the GFD group. However, in contrast to the aforementioned study by Nikniaz et al. [13], we could not find negative PRAL values in the GFD group (which would have indicated alkalizing properties). A comparison with the Iranian study appears generally difficult because the authors did not report the energy intake or protein intake of their cohort.

DAL scores were based on total nutrient intake, and the intergroup differences in magnesium, potassium and protein intakes in our sample could explain the lower PRAL values in our GFD group. Notably, there was an energy difference of almost 300 kcal/d between both groups. Total energy intake has been positively associated with PRAL scores in numerous studies and could also contribute to our findings [15,16].

Crude DAL scores were significantly lower in the GFD group, yet both employed NEAP scores (NEAP_F_ and NEAP_R_) were approximately 50 mEq/d and thus generally comparable to Western diets that produce between 50 and 75 mEq/d [15,28].

The health effects of gluten-free diets (outside the context of CD) are controversial. Many individuals adopt gluten-free diets for their perceived health benefits (and no apparent medical reasons), although nutritional adequacy is subject to ongoing research [11,29]. Recently, the potentially lower acidogenic potential of gluten-free diets has been discussed in social media and blogs [30]. We argue, however, that it is too early to make a definite statement with regard to this point. Our data do not support the strong alkalizing potential of gluten-free diets as reported by Nikniaz et al. [13]. Instead, we argue that the results from the Iranian cohort were potentially the result of other special dietary features (e.g., the low meat intake as suggested by the authors) in their GFD cohort. Unfortunately, a definite statement may not be made due to the lack of reported nutrient intake data in the Iranian cohort.

Our data do not suggest that a GFD has alkalizing properties. Comparably low PRAL scores (as shown by Nikniaz et al. [13]) are usually only observed in plant-based cohorts (e.g., in individuals consuming strict vegetarian and vegan diets) [31,32,33,34]. While a high acidogenic diet has been associated with numerous health repercussions [35,36,37,38], we doubt that this may be buffered with a GFD.

It appears too early at this stage to draw definite conclusions. With only two studies available in this particular field, additional research is urgently warranted. Larger trials supported by food group analyses could help to disentangle the overall complex picture of DAL in gluten-free diets.

Our study has several strengths and limitations that warrant a thorough discussion. As for the strengths, our analysis relies on a nationally representative dataset (from the National Health and Nutrition Examination Surveys). We explored a field that received comparably little attention in the past and pave the way for new research. Unlike Nikniaz et al. [13], we reported nutrient intake and other dietary measures to allow for a more comprehensive picture of DAL in gluten-free diets.

In the meantime, our analysis has several deficiencies worth mentioning. Despite the large, population-based sample, only a modest number of individuals reported a GFD (*n* = 187). Appending additional NHANES cycles, however, would not have solved the problem since the necessary variables were removed after 2014. The absence of food group analyses and various unreliable weighted proportions are additional weaknesses of this analysis. Ultimately, GFD status was self-reported, which could introduce bias and other related problems.

## 5. Conclusions

The present study explored DAL scores in a cohort of individuals consuming a GFD and contrasted the results to the general population denying a special diet. A GFD was associated with significantly lower crude DAL scores. Even after adjustments for confounders, the results remain significant. Yet, our study could not confirm a previous study that reported alkalizing properties of a GFD. DAL scores in our cohort were significantly higher as compared to those found by Nikniaz et al. [13]. Thus, it remains debatable whether the low PRAL scores in the previous GFD study were the result of replacing gluten-rich with gluten-free grains or whether they were due to other dietary features (e.g., a substantially lower meat intake). The effects of gluten-free diets on DAL are poorly understood, and additional research is thus warranted.

## Figures and Tables

**Figure 1 nutrients-14-03067-f001:**
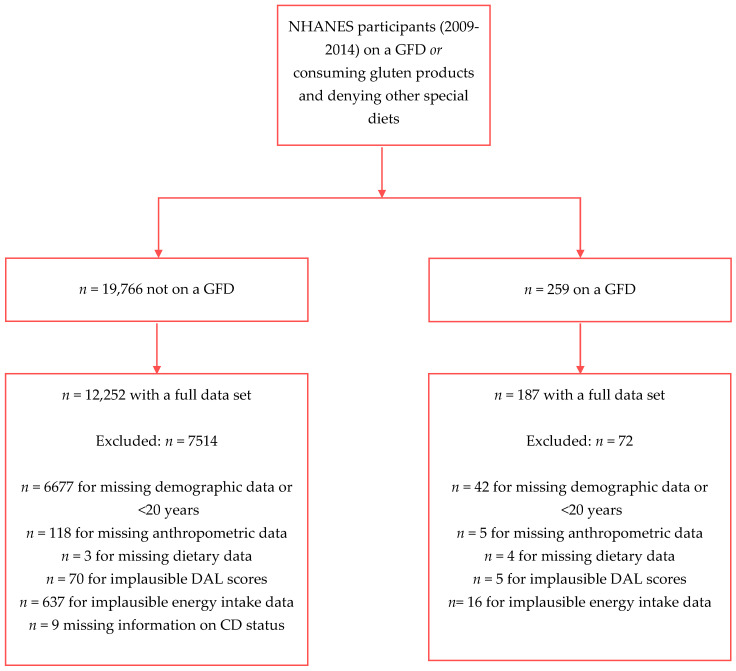
Participant inclusion flow chart.

**Figure 2 nutrients-14-03067-f002:**
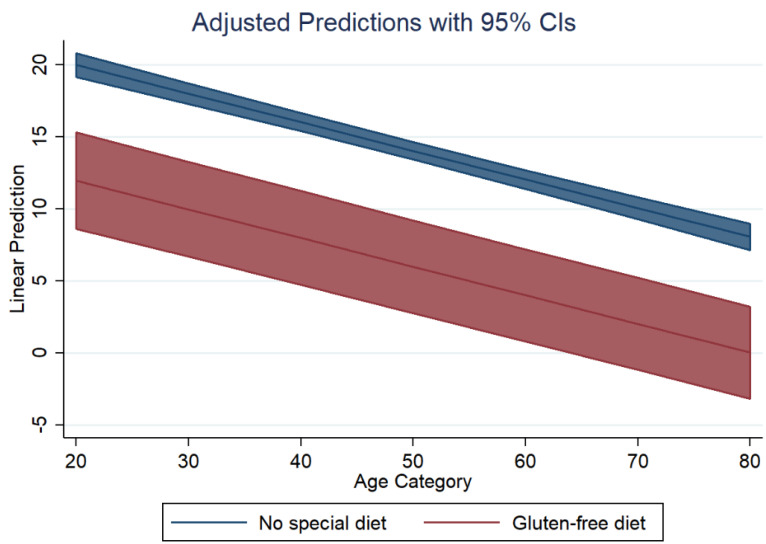
Dietary acid load in both groups: adjusted predictions for PRAL_R_.

**Figure 3 nutrients-14-03067-f003:**
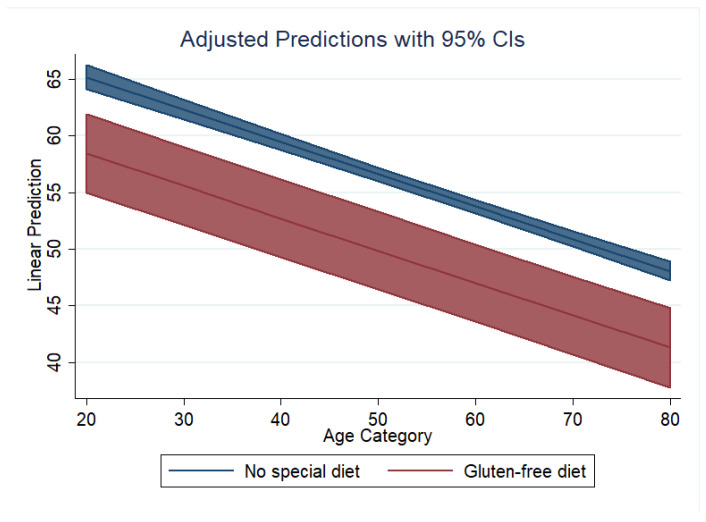
Dietary acid load in both groups: adjusted predictions for NEAP_F_.

**Table 1 nutrients-14-03067-t001:** PRAL values of selected gluten-containing and gluten-free foods: an overview.

Gluten-Containing Foods	PRAL Value per 100 g	Gluten-Free Foods	PRAL Value per 100 g
Rye	+4.4 mEq/d	Quinoa	+2.4 mEq/d
Wheat	+8.2 mEq/d	Millet	+8.6 mEq/d
Spelt	+8.8 mEq/d	Corn	+3.8 mEq/d
Barley	+5 mEq/d	Amaranth	+7.5 mEq/d

The 1:1 exchange of 100 g of gluten-containing grains with gluten-free (pseudo-) grains does hypothetically not lead to significant reductions in DAL. d = day Modified from: https://inaturally.com.au/wp-content/uploads/2020/04/The-PRAL-Table.pdf (accessed on 24 July 2022).

**Table 2 nutrients-14-03067-t002:** Sample characteristics: demographic and anthropometric data.

	No Special Diet (*n* = 12,252)	Gluten-Free Diet (*n* = 187)	*p*-Value
**Age (years)**	46.79 (0.37)	47.87 (2.01)	*p* = 0.595
**Sex**			*p* = 0.005
- Males (*n*(%))	6130 (50.02)	73 (34.29) ^a^	
- Females (*n*(%))	6122 (49.98)	114 (65.71) ^a^	
**Body Mass Index (kg/m²)**	28.43 (0.11)	27.04 (0.41)	*p* = 0.002
**Race/Ethnicity**			*p* = 0.441
- Mexican American	1765 (8.78)	14 (4.94) ^b^	
- Other Hispanic	1151 (5.59)	17 (4.34) ^b^	
- Non-Hispanic White	5433 (67.17)	83 (72.10)	
- >Non-Hispanic Black	2533 (11.17)	43 (9.99)	
- Other Race	1370 (7.29)	30 (8.63) ^b^	
**Marital Status**			*p* = 0.48
- Married/living with partner	7238 (61.88)	104 (56.30)	
- Widowed/divorced/separated	2590 (18.08)	42 (17.86)	
- Never married	2424 (20.05)	41 (25.84)	
**Education Level**			*p* = 0.009
- Less than 9th grade	1123 (5.02)	9 (2.33) ^a,b^	
- 9–11th Grade	1815 (11.57)	15 (7.44) ^b^	
- High School Grad/GED	2819 (22.01)	33 (13.60) ^b^	
- Some College or AA degree	3599 (31.90)	52 (28.15)	
- College Graduate or above	2896 (29.50)	78 (48.48) ^a^	

Values for continuous variables expressed as estimated mean and standard error in parentheses. Table displays weighted proportions for categorical variables. ^a^ indicates significant differences in the weighted proportions; ^b^ indicates unreliable proportions.

**Table 3 nutrients-14-03067-t003:** Sample characteristics: energy, macro- and micronutrient intake.

	No Special Diet (*n* = 12,252)	Gluten-Free Diet (*n* = 187)	*p*-Value
Calories (kcal/day^)^	2180.89 (8.72)	1890.42 (70.26)	*p* = <0.001
Carbohydrate (g/1000 kcal)	122.40 (0.32)	112.80 (3.45)	*p* = 0.008
Carbohydrate total (g/day)	263.87 (1.22)	213.77 (10.31)	*p* = <0.001
Protein (g/1000 kcal)	38.69 (0.23)	42.60 (1.77)	*p* = 0.030
Protein total (g/day)	82.82 (0.47)	78.61 (3.89)	*p* = 0.287
Fat (g/1000 kcal)	37.17 (0.12)	39.96 (1.54)	*p* = 0.072
Fat total (g/day)	82.29 (0.46)	77.58 (4.56)	*p* =0.306
Potassium (mg/1000 kcal)	1300.32 (8.38)	1567.17 (36.07)	*p* = <0.001
Potassium total (mg/day)	2725.46 (15.10)	2892.29 (113.30)	*p* = 0.145
Magnesium (mg/1000 kcal)	145.28 (0.94)	183.4 (5.43)	*p* = <0.001
Magnesium total (mg/day)	305.70 (1.79)	339.84 (13.25)	*p* = 0.013
Phosphorus (mg/1000 kcal)	655.85 (2.89)	702.29 (21.98)	*p* = 0.038
Phosphorus (mg/day)	1407.97 (6.46)	1307.08 (56.21)	*p* = 0.076
Calcium (mg/1000 kcal)	462.48 (2.91)	492.58 (26.93)	*p* = 0.273
Calcium (mg/day)	985.33 (6.02)	903.89 (51.49)	*p* = 0.128
Sodium (mg/1000 kcal)	1680.86 (7.20)	1638.60 (46.59)	*p* = 0.369
Sodium (mg/day)	3599.16 (16.26)	3058.62 (146.65)	*p* = 0.001

Values expressed as estimated mean and standard error in parentheses.

**Table 4 nutrients-14-03067-t004:** Sample characteristics: dietary acid load scores.

	No Special Diet (*n* = 12,252)	Gluten-Free Diet (*n* = 187)	*p*-Value
PRAL (mEq/day)	14.68 (0.34)	5.56 (2.39)	*p* = <0.001
NEAP_R_ (mEq/day)	59.92 (0.36)	48.67 (2.51)	*p* = <0.001
NEAP_F_ (mEq/day)	57.50 (0.37)	51.14 (2.50)	*p* = 0.016

Values expressed as estimated mean and standard error in parentheses.

**Table 5 nutrients-14-03067-t005:** Weighted linear regression estimates of predictor variables for outcome variable (PRAL_R_): model 1 (left columns) and model 2 (right columns).

PRAL_R_	Coefficient	95% CI	*p*	Coefficient	95% CI	*p*
	Model 1			Model 2		
**Diet**						
No special diet	-	-		-		
Gluten-free diet	−5.11	(−10.14, −0.08)	0.047	−8.02	(−11.2, −4.84)	<0.001
**Gender**						
Male	-	-		-	-	
Female	−3.24	(−4.24, −2.25)	<0.001	0.06	(−0.75, 0.88)	0.878
**Body Mass Index**	0.27	(0.18, 0.36)	<0.001	0.22	(0.14, 0.31)	<0.001
**Age**	−0.19	(−0.21, −0.16)	<0.001	−0.20	(−0.22, −0.18)	<0.001
**Race/Ethnicity**						
Mexican American	4.12	(2.43, 5.80)	<0.001	1.92	(0.37, 3.48)	0.016
Other Hispanic	2.88	(1.25, 4.50)	0.001	0.24	(−1.12, 1.61)	0.722
Non-Hispanic White	-			-		
Non-Hispanic Black	3.68	(2.32, 5.04)	<0.001	2.64	(1.09, 4.18)	<0.001
Other Race—Including Multiracial	1.48	(−0.15, 3.10)	0.074	−0.70	(−2.05, 0.65)	0.305
**Energy intake (kcal/day)**	0.009	(0.009, 0.01)	<0.001	−0.005	(−0.006, −0.004)	<0.001
**Protein intake (g/day)**				0.47	(0.44, 0.49)	<0.001

Coefficients are displayed with their 95% confidence intervals and *p*-value. The symbol “-” indicates the reference category. *p* = *p*-value.

**Table 6 nutrients-14-03067-t006:** Weighted linear regression estimates of predictor variables for outcome variable (NEAPF): model 1 (left columns) and model 2 (right columns).

NEAP_F_	Coefficient	95% CI	*p*	Coefficient	95% CI	*p*
	Model 1			Model 2		
**Diet**						
No special diet	-	-		-		
Gluten-free diet	−4.25	(−9.23, 0.72)	0.092	−6.74	(−10.12, −3.29)	<0.001
**Gender**						
Male	-	-		-	-	
Female	−4.57	(−5.44, −3.70)	<0.001	−1.74	(−2.59, −0.89)	<0.001
**Body Mass Index**	0.348	(0.25, 0.44)	<0.001	0.31	(0.21, 0.40)	<0.001
**Age**	−0.27	(−0.30, −0.25)	<0.001	−0.29	(−0.31, −0.26)	<0.001
**Race/Ethnicity**						
Mexican American	2.40	(0.29, 4.50)	0.026	0.52	(−1.65, 2.69)	0.630
Other Hispanic	4.23	(2.36, 6.10)	<0.001	1.98	(0.07, 3.90)	0.043
Non-Hispanic White	-			-		
Non-Hispanic Black	8.38	(6.93, 9.82)	<0.001	9.01	(7.76, 10.27)	<0.001
Other Race—Including Multiracial	3.17	(1.28, 5.07)	0.002	1.32	(−0.33, 2.97)	0.116
**Energy intake (kcal/day)**	0.001	(0.001, 0.002)	<0.001	−0.01	(−0.01, −0.01)	<0.001
**Protein intake (g/day)**				0.40	(0.37, 0.43)	<0.001

Coefficients are displayed with their 95% confidence intervals and *p*-value. The symbol “-” indicates the reference category. *p* = *p*-value.

## Data Availability

Data are publicly available online (https://wwwn.cdc.gov/nchs/nhanes/Default.aspx; accessed on 2 July 2022). The datasets used and analyzed during the current study are available from the corresponding author upon reasonable request.

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
