# Peer review of "Dietary Acid Load in Gluten-Free Diets: Results from a Cross-Sectional Study"

_nutrients, 2022, doi:10.3390/nu14153067_

Round 1

Reviewer 1 Report

I am skeptical about this manuscript. The study concept is translucent to me. I believe, the authors wanted to compare the Dietary Acid Load between gluten-free and gluten-containing diet. However, this is not clearly mentioned in the manuscript. I have some comments below:

Comments

Comment#1: I am not fully sure about the concept of the study. Probably, the authors want to mention that healthy individuals are adopting GFD, that is not recommended and not healthy for them. If I understood well, I suggest the authors to mention this openly. The same has not been explained in the abstract as well.

Comment#2: The introduction is weak. It does not effectively explain the purpose and need of the study (specially the last part).

Comment#3: Almost this whole paragraph does not add any information related to the study. Line no 84-89

Comment#4: Please state following points clearly here:

 type of participants, study duration, inclusion/exclusion criteria, location of study site. Line no. 90-93

Comment#5: Please give more detail about the participants or cite the related table/figure. Line no. 165-166

Comment#6: what is this ‘no special diet’?

Comment#7: I did not understand the significance of ‘NHANES’ in the study title.

Comment#8: Please mention the full form of NHANES. Line no.  78

Comment#9: Gluten-free diet is used as singular noun. I am not sure why the author used it as a plural at some occasions and then again used it as singular form in later sections.

Comment#10: Line no 35-36 are not transparent.

Comment#11: What this number denotes 1623? Line no. 95

Comment#12: What is this error? Line no. 117

Comment#13: What is ‘interview component’? Line no.121

Comment#14: Reference style does not follow the journal’s recommended citation style.

Author Response

Dear Reviewer,

We would like to thank you very much for careful and thorough reading of this manuscript and for the thoughtful comments and constructive suggestions, which help us to improve the quality of this article. Please kindly find our response below. All requested changes have been clearly marked in yellow and blue color. We appreciate your input, your advice and the fast peer review. Thank you!

Reviewer 2 Report

The manuscript “Dietary Acid Load in Gluten-Free Diets: Results from a Cross- Sectional Study (NHANES)” presents the main results of a study investigating, on a sample of US population, if the gluten free diet can have a lower acidogenic potential. The study is of interest for the scientific community, however it requires revision before publication.

At first sight, it emerged that references within the text were wrongly indicated. The reference list also does not match Journal guidelines. So, please check them and amend the manuscript accordingly.

INTRODUCTION

Check Table layout. The indication “Table 1” below the table is not necessary.

MATERIALS AND METHODS

Lines 78-88: these lines are more suitable for an introduction than for the “Materials and Methods” section.

Line 97: I suggest defining what “NEAPR and NEAPF” stand for. The same apply to all acronyms (e.g. OAest, etc.). Please, explain the acronym the first time it appears, and then use the acronym.

Line 117: Please, check the reference.

RESULTS

Tables: Please, check Journal guidelines for Table captions and legends.

Table 6: I suggest the Table contains the information Model 1 and Model 2. Otherwise it is not clear at a first glance.

Line 279: check spelling “meat” and not “meet”.

Author Response

(The authors gave the same response as above.)

Reviewer 3 Report

The present study explored DAL scores in a cohort of individuals consuming a GFD and contrasted the results to the general population denying a special diet. GFD were associated with significantly lower crude DAL scores.

Even after adjustments for confounders, the results remained significant. Yet, our study could not confirm a previous study that reported alkalizing properties of a GFD. DAL scores in our cohort were significantly higher as compared to Nikniaz et al.

Thus, it remains debatable whether the low PRAL scores in the previous GFD study were the result of replacing gluten-rich with gluten free grains or whether they were due to other dietary features (e.g. a substantially lower meet intake).

The effects of GFDs on DAL are poorly understood and additional  research is thus warranted.

Author Response

Dear Reviewer,

We would like to thank you very much for careful and thorough reading of this manuscript and for the very positive comments and constructive suggestions, which help us to improve the quality of this article. Please kindly find our response below. All requested changes have been clearly marked in yellow and blue color. We appreciate your input, your advice and the fast peer review. Thank you!

Round 2

Reviewer 1 Report

I appreciate the authors for their polite behavior and their way of presenting the responses. I confront that I am also one of those who do not know much about NHANES. I am happy with almost every response.

In response to comment#4, the authors' reply is commendable. If I am right, now it is clear that the authors have collected patients’ data from NHANES modules between 2009-2014. That means the study was a retrospective study. Secondly, I am still not sure when the study was conducted? I mean when the collected data were analyzed? I am sure It was analyzed after 2014 and somewhere between 2021-2022. Please mention this in the manuscript.

Author Response

Dear Reviewer,

We would like to thank you again very much for careful and thorough reading of our manuscript and for your positive comments, which help us to improve the quality of this article. Please kindly find our response in the attachment.

Sincerely,

The authors
